# A Physiological Approach to Vocalization and Expanding Spoken Language for Adolescents with Selective Mutism

**DOI:** 10.3390/bs15081013

**Published:** 2025-07-25

**Authors:** Evelyn R. Klein, Cesar E. Ruiz

**Affiliations:** Department of Communication Sciences and Disorders, La Salle University, Philadelphia, PA 19141, USA; ruiz@lasalle.edu

**Keywords:** selective mutism, anxiety disorder, communication, pragmatic language, voice, vocal control, treatment, speech, fear, outreach

## Abstract

Selective Mutism (SM) is a childhood anxiety disorder characterized by the persistent inability to speak in specific social settings while being able to speak freely in more comfortable environments, such as at home with family. This condition often leads to significant impairments in social, academic, and occupational functions. This article presents a novel treatment methodology that integrates the physiology of vocal production with pragmatic language instruction through teletherapy, administered to two adolescents diagnosed with selective mutism (SM). The frequency of speaking on the Selective Mutism Questionnaire increased from 35% to 86% and from 25% to 55% for the two children. Pragmatic language skills on the Social Communication Skills: Pragmatics Checklist improved from 47% to 96% and 13% to 40% after treatment. It is crucial to emphasize vocal control for speech initiation and pragmatic language for verbal expression. Detailed strategies, specific activities, and treatment outcomes are provided.

## 1. Introduction

Selective Mutism (SM) is a complex childhood anxiety disorder marked by a persistent inability to speak in specific social settings where speech is expected, despite being able to speak freely in more comfortable environments, such as at home ([1]; [6]; [33]). This anxiety-driven silence affects communication skills, including initiating speech, organizing language, and effectively expressing thoughts ([27]). [33] ([33]) found that children with SM often fear making mistakes, using incorrect or inappropriate words, mispronunciations, or having their voice sound unusual. Such disruptions can lead to significant impairments in social, academic, and daily functioning, emphasizing the critical need for early detection and intervention.

SM affects approximately 0.1% to 2.2% of the population, depending on the diagnostic criteria and sample demographics ([19]). Symptoms typically emerge upon school entry, when verbal communication becomes more socially expected. Females are diagnosed more frequently than males, with ratios ranging from 1.5:1 to 2.6:1 ([3]; [8]), although the reasons behind this gender difference remain unclear.

In a recent systematic literature review of long-term outcomes of selective mutism ([12]), 2432 papers were screened. Ultimately, nine clinical cohorts and two case-control studies were reviewed. The results found that most children with SM recovered during adolescence, but that anxiety disorders, especially social anxiety, were commonly found later in life, with 6% to 54% experiencing an anxiety disorder. Early detection and therapy are important to prevent the emergence of persistent symptoms and disorders. The rates of recovery from SM ranged from 46% to 100%. The majority (78%) of the cases made substantial gains in communication.

The etiology of SM is multifactorial, stemming from the interplay of genetic predispositions, environmental stressors, and individual psychological traits ([23]). Children with a family history of anxiety are at a greater risk. Environmental factors, such as parental anxiety, family interactions, and exposure to trauma, contribute to the onset and persistence of these disorders ([32]). Psychologically, traits like extreme shyness, social inhibition, and sensory sensitivity increase susceptibility ([17]; [16]). These interrelated influences underscore the need for a comprehensive, multidisciplinary approach to assessment and treatment.

The diagnosis of SM involves a structured, in-depth evaluation by qualified professionals such as psychiatrists, psychologists, and pediatricians ([18]; [6]; [15]). The process typically includes clinical interviews, cross-setting observations and standardized assessments ([22]; [10]). The diagnostic criteria specify that the child consistently fails to speak in particular social contexts despite speaking in others, and this behavior significantly interferes with functioning. Speech-language pathologists (SLPs) can evaluate a child’s speech and language capabilities to identify any communication challenges that may be influencing selective mutism or to diagnose other existing communication disorders.

It is crucial to distinguish SM from other conditions, such as stuttering and autism spectrum disorder (ASD). However, research indicates that there are overlapping features between SM and ASD, especially in areas of social communication and anxiety, and it is possible for an individual to have SM and ASD concomitantly ([30]). Although distinct, children with ASD may exhibit selective silence due to similar social challenges. Therefore, clinicians must assess shared behaviors, such as reduced joint attention, impaired social reciprocity, and restrictive-repetitive behaviors, to ensure accurate diagnosis and effective intervention planning ([5]; [31]).

The treatment of SM requires a multifaceted, personalized strategy targeting both anxiety and communication barriers. Cognitive-Behavioral Therapy (CBT), particularly through gradual exposure and cognitive restructuring, has shown strong effectiveness. Techniques such as systematic desensitization and positive reinforcement have also proven beneficial ([34]; [17]). Other behavioral strategies, such as exposure, shaping, and stimulus fading, have also been used successfully in treatment ([20]).

### 1.1. Treatment Considerations

Speech-language therapy plays a vital role in enhancing vocal control and demystifying the speech process through psychoeducation ([26]). This educational approach clarifies the physiological production of sound, enabling children to build expressive skills once vocalization has been initiated. Beyond verbal output, therapy aims to foster self-expression and emotional release through the voice. Neuroimaging studies have linked vocal control deficits in anxiety to structural changes in brain areas, such as the anterior cingulate and inferior frontal gyrus ([28]). Accordingly, [24] ([24], [25]) and [26] ([26]) advocate vocal control exercises, including humming, whispering, and voice initiation, as effective tools for overcoming difficulties in initiating speech. To measure laryngeal tension and guide intervention, surface electromyography (sEMG) has been used to objectively track the muscular constraints on the voice.

The development of social pragmatic language is another key treatment focus. This includes using language for various communicative intents (e.g., requesting clarification, apologizing, or providing information) and adapting communication based on the context and listener. It also involves mastering conversational rules, such as turn-taking and topic maintenance. Limited practice in these areas, especially with decontextualized language (discussing thoughts and ideas that are not present in the immediate environment), may impede effective communication.

In more severe cases, pharmacological treatment may complement the behavioral interventions. Selective serotonin reuptake inhibitors (SSRIs) have been effective in reducing anxiety symptoms in children with SM, helping them to initially engage in therapy ([14]). Medication is generally reserved for situations in which anxiety significantly obstructs therapeutic progress ([14]).

Integrated treatment plans combining behavioral, psychological, speech-language, and pharmacological approaches tailored to each child’s needs yield the best outcomes ([21]). This article discusses how anxiety affects vocal mechanisms and limits pragmatic language development in social settings. The ECHO Program ([26]) supports vocal control and enhances spoken language in individuals with SM.

### 1.2. Theoretical Framework

The foundation of this therapeutic work is based on the ARC model ([11]), which addresses the emotional barriers to speaking. ARC stands for anxiety tolerance, rescue reduction, and communication confidence. Figure 1 illustrates an arc representing variations in anxiety experienced in daily situations. Anxious individuals may avoid anxiety-inducing environments to reduce their stress levels. Common emotional responses include fear, self-doubt, and perceived stress. The fear response of freezing, mediated by the autonomic nervous system, is understood to affect motor activity, such as the movement of body parts and motor activity required for vocalization ([13]). The freezing response might be associated with reduced motor ability to vocalize in selected situations for children with SM. When children gain experience and support in increasingly challenging communicative situations, they can stop avoiding and begin managing their anxiety. With consistent targeted practice, confidence in speaking can increase over time, and communication becomes more comfortable. This framework provides the basis for the vocal control-pragmatic language approach to treating SM.

In situations that provoke anxiety, speech production is often inhibited. Physiologically, this can be explained by the observations of many children formally assessed ([25]; [10]). While they express a strong desire to speak, they report feeling physically incapable of doing so. Speech begins with vocalization, which requires a steady stream of air from the lungs to vibrate the vocal cords. The resulting sound is then shaped by the articulators in the mouth. For example, the lips come together to form the “m” sound, the mouth opens for “ah,” and then returns to “m” to create the word “mom.”

The initiation of voluntary voice is regulated by the anterior cingulate cortex (ACC), a region of the limbic system that is responsible for emotional regulation. The periaqueductal gray (PAG) supports the coordination of stress and threat responses, helping manage behavior under pressure. Together, these areas are central to processing fear, anxiety, and emotional reactivity ([2]; [29]). When anxiety disrupts the functioning of these neural systems, the ability to produce voice—and, therefore, speech—can be compromised ([28]).

Understanding this physiological interference, the goal of this study was to assess and treat difficulties with voice initiation in individuals with SM ([25]). Once vocalization becomes possible, intervention can shift toward building speech and expanding pragmatic language for everyday communication purposes. This therapeutic process is supported by two case studies that demonstrate the use of teletherapy to improve vocal control and pragmatic language development ([26]).

Gaining control of the voice for speech initiation is essential. Research has shown that children with SM have substantial vocal tension when speech is expected from them, even before they say anything. This was measured using surface electromyography ([25]). Interactive activities organized in a hierarchical sequence, designed to teach how speech sounds are produced and enhance intentional voice initiation and control, were implemented. The individuals gained foundational knowledge of how speech is generated and how sounds are combined in sequence to form words. This process is integral to helping older children understand the mysteries of speaking.

## 2. Materials and Methods

Within this case study design, the two participants are described along with the measures, procedures, and treatment progress.

### 2.1. Participants

The two cases presented in this study were examined by authors who specialize in SM. Parents contacted specialists after reviewing the Selective Mutism Association website. After the family made initial contact with the licensed speech-language pathologists and psychologists (one of the authors is both a speech-language pathologist and psychologist), parents and teens received information about the vocal control-pragmatic language treatment approach, and they decided to start treatment.

#### 2.1.1. Case 1: GB

*Age at start of treatment*—12 years; 8 months

*Gender and Race*—Female, Caucasian

*Birth Information*—Born full-term (38 weeks) in good physical health, 6 lbs.12 oz.

*Schooling*—Attended parochial elementary school, 6th-grade start of current treatment

*Duration and Treatment*—35, 1 h sessions, for a duration of 16 months from 6th grade to 8th grade, graduating middle school.

*Presenting Problem and Reason for Referral*—SM with spontaneous language challenges. 

*Diagnosis*—Selective Mutism (ICD-10 diagnostic code—F94.0).

*Family Background*—GB lived with her biological mother, father and one sister (4 years older). There was a history of social anxiety in her immediate family. GB indicated that her parents remained involved in all aspects of her life and that she was very close with them. Her father was a corporate executive, and her mother was a teacher and a full-time stay-at-home mother.

*Social Background*—Behaviorally, GB preferred to play alone as a young child. She did not have many friends and was quiet during preschool. Screening with the SCARED ([4]) indicated concerns about generalized anxiety and social anxiety. GB’s social life consisted of a few friends at school and spending time with her immediate family. She also engaged in weekly extracurricular activity of dance, where she received formal training for recitals and dance competitions. GB indicated that she felt nervous at times and worried about being as good as the other children. Although she initiated speech with her parents, she had difficulty in describing her feelings. When at a restaurant with her family, she would order using one or two words when her parents prompted her.

*Developmental History*—Developmental milestones were achieved at age-appropriate times. Hearing and vision tests were within normal limits. No developmental concerns were noted.

*Medical History*—GB had a history of ear infections, allergies, and influenza. According to her parents, she was an anxious child who primarily interacted with her family. To relieve her anxiety, Zoloft was prescribed for a period of 18 months. between the 2nd and 3rd grades. Parents reported that the medication only moderately improved inhibition and decreased sensory sensitivity to touch and sound. The medication was discontinued due to weight gain and lack of substantial improvement. Otherwise, the GB was considered healthy.

*Educational Information*—GB received high grades in all subject areas, according to the parents. During her early school years, her parents noted that she displayed significant effort to master academic concepts. GB became a serious student and spent much time studying to obtain good grades. She was not satisfied with any grade lower than 95. This intensity continued throughout her treatment.

*History of SM and Prior Treatments*—Evidence of SM was present in preschool at 4 years of age, but was not formally recognized until kindergarten. According to the parents, GB received 6 years of therapy related to SM with six different practitioners specializing in SM prior to seeking the current services that follow within this case report. GB began treatment in kindergarten after being diagnosed with SM (ICD-10 code F94.0). Speech therapy focused on speaking and reading, and occupational therapy focused on sensory needs with auditory integration to improve listening and use of speech. In first grade, GB spoke in a full voice and answered questions but was mute with classmates. She made gains toward the end of the school year by whispering to several friends in her class. At home, she was fully conversant. Before 2nd grade, she attended an intensive program for SM and received over 40 h of therapy. Although the GB worked hard, there was no major breakthrough. She was unable to speak in front of her class in second grade but did speak socially to a few friends. In 3rd and 4th grades, she expanded her talking circle to about 12 girls in her class. She was unable to speak to her teacher or other adults at school. She used a speaking buddy to answer questions for her teacher. In the 4th and 5th grades, she made video recordings at home and played them for the teacher to complete speaking assignments. Video sharing helped her connect with her peers. In 5th grade, she began to answer her teacher with an audible voice. The prospect of greeting people, talking to students that she had not spoken to, and speaking to the full class appeared to be too much for her to contemplate at that time. Her parents decided to stop focusing on SM and wait. Parents contacted our practice regarding treatment when GB was nearing the end of 6th grade.

*Psychoeducational Testing*—At the end of her current treatment for SM and in preparation for high school, GB received a comprehensive evaluation to provide information about her neurocognitive, academic, and social-emotional functioning. According to the evaluator (a licensed psychologist), GB made substantial progress in managing her symptoms of SM and no longer met the criteria for SM. However, she continued to experience anxiety and more compulsive behaviors. GB exhibited strengths in verbal, visual-spatial, and working memory, successive information processing, mathematics, and fluid reasoning. Processing speed, planning, attention, simultaneous information processing, visual-motor integration, visual perception, fine motor coordination, and reading and writing skills were considered age-appropriate. Her WISC-V full-scale IQ was 122 ( 93rd percentile rank). She was diagnosed with generalized anxiety disorder (ICD-10 code F41.1) and a suggestion to rule out obsessive-compulsive disorder in the future, should the need to complete specific routines and rituals in a particular way, escalate.

#### 2.1.2. Case 2: KD

*Age at start of treatment—*16 years; 10 months

*Gender and Race*—Male, Caucasian

*Birth Information*—Born full-term (39 weeks) in good physical health, 7 lbs.14 oz.

*Schooling*—Attended public high school—in 11th grade at start of treatment

*Duration and Treatment*—39, 1-h sessions, for a duration of 12 months from 11th to 12th grade, graduating high school.

*Diagnosis—*SM (ICD-10 diagnostic code—F94.0) with autism spectrum disorder (mild) (ICD-10 diagnostic code—F84.0), and communication challenges.

*Family Background*—KD lived with his biological parents and his 18-year-old sister, who is in college. There was a history of anxiety in the immediate family and high-functioning autism on his father’s side. His mother was very involved in his life and felt that the school did not provide adequate support for him. She stated that KD often felt frightened, worried about his performance, and disliked being away from his family. He was shy and uncomfortable reading aloud or speaking in front of others and avoided places with unfamiliar people.

*Social Background*—KD showed little interest in social interactions and preferred online gaming with a few peers. He rarely initiated or engaged in communication and felt no need for casual conversations. At home, he communicated minimally and only when necessary. He lacked affect and did not view his mutism as problematic, often responding with brief responses. However, he demonstrated affection for his dog during therapy sessions.

*Developmental History*—KD met developmental milestones on time. He passed hearing and vision tests without any issues but had a history of picky eating. Early development included aloofness, compulsive behaviors, meltdowns at 2 years of age, severe separation anxiety, poor eye contact, and withdrawal. At preschool, he struggled with directions and conversations, showing impaired pragmatic language skills, lack of joint attention, imitation, pretend play, and social smiling.

*Medical History*—KD experienced seasonal allergies and frequent colds during his childhood. According to KD’s mother, there were concerns regarding head injuries due to a fall while ice skating and another incident involving hitting his head against a door. Gastrointestinal issues were noted in his youth, for which he took laxatives for two years. KD had a history of tics and sensory sensitivities, particularly related to auditory stimuli and bright lights. During this SM treatment, after turning 17 years old, KD’s bone age was found to be two years delayed, resulting in a smaller stature and low weight (5 feet 3 inches, weighing 107 lbs.). Previous treatments for anxiety included antidepressants, which were ineffective. All medications, including a trial of 30 mg of Prozac, were discontinued.

*Educational Information*—KD disliked school but never refused to attend. He completed his schoolwork because it was expected of him. He followed the rules but was resistant to change. KD performed well in science and math when he was younger, but as the verbal and conceptual material became more abstract, he lacked interest. Memory, math skills, vocabulary, and executive functioning were measured to be above average for his age. He showed an interest in the computer networking field and attended a high school for technology in network operating systems and security. He was accepted into a college specializing in technology with plans for a career in cybersecurity. He intended to live on campus, which is about two hours away from his home.

*Psychoeducational Testing*—KD was diagnosed with selective mutism (ICD-10 code F94.0) and mild autism spectrum disorder (ICD-10 code F84.0) at the age of 10 years, during his 5th-grade year. Commencing in 6th grade, KD received gifted support services. His diagnosis of autism was confirmed through observations, interviews, and performance on the ADOS-2, highlighting significant challenges in social and communication skills throughout the assessment. At the age of 16, KD was further diagnosed with tension dysphonia (ICD-10 code R49.8) and social pragmatic communication disorder (ICD-10 code F80.82). At this juncture, evaluations of his intellectual functioning revealed a considerable discrepancy between his verbal and nonverbal reasoning abilities. The WISC-V Composite standard scores were as follows: verbal comprehension = 95 (37th percentile), visual spatial = 132 (98th percentile), fluid reasoning = 126 (96th percentile), working memory = 103 (58th percentile), processing speed = 92 (30th percentile), and full-scale IQ = 109 (73rd percentile). Assessments of social, emotional, and behavioral functioning indicated elevated levels of inhibition, submissiveness, conformity, anxiety, and insecurity, especially in peer interactions.

*History of SM and Prior Treatments*—KD received speech therapy for a lisp and occupational therapy for sensory issues in the 3rd grade. At 13 years of age, he underwent private therapy for handwriting. At 15, he attended an intensive one-week summer camp for SM, where he initially did well but faced increasing challenges once the group ended. He made some progress in speaking to meet his needs in the community, but it did not generalize effectively. KD struggled with the follow-up virtual groups and did not maintain any interactions. Despite his mother’s efforts to get therapeutic support in school, standardized assessments did not show enough deficits for formal treatment. She contacted our practice when KD was in 11th grade.

### 2.2. Measures

Prior to treatment, the initial measures included a thorough case history with a formal diagnosis of SM. Initially, information was gathered from parents using the Social Communication Skills: Pragmatics Checklist ([7]), Selective Mutism Questionnaire ([3]), and Express Selective Mutism Communication Questionnaire ([9]), and copies of previous assessment reports from parents and teachers, as available.

*The Social Communication Skills: Pragmatics Checklist* (SCS-PC) ([7]) is a 45-item checklist that assesses the use of language in a socially appropriate manner for specific social purposes. According to [7] ([7]), children between 3 and 4 years of age use complex language to convey pragmatic language skills. Without these skills, children face significant challenges in literacy, language, and conversational communication. Using a 4-point Likert scale, six areas of social pragmatic language were evaluated: Instrumental (stating needs), Regulatory (giving directions), Personal (expressing feelings), Interactional (engaging in two-way communication with pragmatic speech acts such as initiating and maintaining a conversation), Wants Explained (asking questions for information), and Sharing Knowledge & Imaginations (such as telling a story and relating information). Ratings for each of the 45 items ranged from 0 to 3, with 0 (not present), 1 (uses no words—gestures), 2 (uses 1–3 words), or 3 (uses complex language) for 135 total possible points.

*The Selective Mutism Questionnaire* (SMQ) ([3]) is a standardized instrument for assessing speaking behavior at school, at home, and in social situations outside school. The items are rated by parent(s)/caregiver(s) to determine the child’s frequency of speech in various situations. Using a 4-point Likert scale from 0 (never speaks), 1 (seldom speaks), 2 (often speaks), and 3 (always speaks), speaking situations are rated to determine the likelihood of a child displaying SM. Higher ratings indicate a higher frequency of speech. A total of 51 points is possible given the 17 speaking situations. The coefficients of internal consistency ranged from 0.65 to 0.91. The total reliability was strong, with a scale coefficient of r = 0.78 ([3]).

*The Express Selective Mutism Communication Questionnaire* (SMCQ) ([9]) is a questionnaire completed by parent(s)/caregiver(s) or administered in an interview format. The components included background information related to diagnoses, symptoms, and treatment. Secured information also includes temperament, anxieties, and the recall of helpful methods that encourage communication. There is also a matrix that helps evaluate how the child responds to and initiates communication with a variety of people, including family, friends, peers, neighbors, clerks, servers, doctors, teachers, and others at home, in school, and in public settings. Comparisons can also be made regarding communication when parents or others are present. Information and details about a child’s social world can be readily obtained ([9]).

### 2.3. Design

This retrospective case study explores the effectiveness of a vocal control/pragmatic language treatment approach for selective mutism in two adolescents. Both participants received treatment over 12 to 16 months, demonstrating progress with different treatment trajectories within a teletherapy framework. First-person accounts, including patient and family interviews and behavioral measures with treatment, are presented.

### 2.4. Procedures

The purpose of the treatment was to help individuals transition from selective mutism to verbal communication in different settings and with different people. The following section outlines this process and provides specific treatment for two teenagers. A physiological treatment approach was used to support vocalization and expand spoken language. The seven guiding principles for the therapist included: (1) assuming a more passive demeanor with reduced eye contact; (2) focusing on the materials initially instead of looking directly at the child; (3) not calling attention to talking or not talking; (4) not pressuring the child to speak; (5) accepting any type of communication, whether nonverbal or whispering, at this early stage; (6) waiting 5 s for a reply and if no reply, asking again and waiting, or changing the question to a choice question giving two or three options, and then asking the question as a yes-no choice if needed; and (7) changing only one thing at a time with consideration to the activity, the people, or the setting.

To begin, the clients made an informed decision based on learning about the physiological approach to vocalization and expansion of spoken language ([26]). Plans for treatment were initially shared with the parents. If they thought the program could be of value to their child, a follow-up meeting was held with the child with SM and their parent(s). During the second meeting, the goals and components of the program were explained. In addition to the selected activities, the progression through the sessions was shared with the children. At the end of this 30 min meeting, the children were asked if they would like to try working with us. They could reply by speaking, writing in the chat, or giving a thumbs up for yes, down for no, or sideways if they were undecided. The individual’s buy-in was a key factor in proceeding with the work.

The two teens in this case study were seen online for teletherapy. The initial meetings progressed from a virtual orientation with the family, including the teen listening, to beginning treatment with their video off and sound off while interacting via chat. Progress included turning the video off and the sound on. Moving forward, the video was on, and the sound was off, but the chat was used to respond. The next progression included video and sound. The trajectory of treatment progress is as follows:

Once the individuals agreed to treatment, the form, *Information About Me*, was completed to learn more from each individual. The form included 15 items to get to know more about the children’s interests, preferred activities, TV shows, movies, apps, computer sites, videos, preferred sports activities, music, favorite foods, information about pets they have or had, and what they like to do in their free time.

Following each session, a structured and consistent process was implemented to document the participant’s progress and ensure active parent engagement. This involved completing individualized SOAP notes (subjective information, objectives of treatment, assessment indicating progress, and plan for upcoming session) and sending personalized emails to parents or caregivers to provide updates and encourage the carry-over of skills at home. Parents were encouraged to respond to these emails with updates, observations, or questions regarding their child’s progress at home. Their input was valuable in tailoring future sessions to meet each child’s individual needs and ensure consistency between the home and clinical environments.

#### 2.4.1. Treatment Module Procedures

The primary goal of treatment was to create meaningful social communication opportunities while developing skills in vocalization, pragmatic language use, and interactive role play. These experiences were designed to help individuals feel more confident and at ease during conversation. For individuals with selective mutism, it is important that therapeutic approaches are practical and realistic, emphasizing the development of vocal control and social pragmatic language skills to support effective communication. Interactive activities and games were used to support skill development and help achieve these goals. The following section describes the vocal control module in more detail.

For vocal control, the individuals learned to initiate and control their voices within seven interactive activities delivered via teletherapy. Each activity included a unique game whereby the child was given remote control access to interact online while playing with the therapist. This process fostered an interactive exchange between the therapist and the client. The goals of each activity are described below.

Sound Off: To increase awareness of nasal, oral, and throat speech sound production; to increase awareness of voicing and distinctive features (nasal/oral/throat, airflow continuation or stop, voice vibration or no vibration) for speech sound production; to increase awareness of articulatory contacts (lips, teeth, palate, tongue, or glottal) for speech sound production; and to identify voicing and distinctive features for speech sound production in words.Pitch Pipe: To demonstrate vocal control for pitch variation spontaneously and on demand and to discuss the concept of high and low voices by demonstrating the differences.Ramp it up: To demonstrate vocal control for loudness variation spontaneously and on demand and to discuss the concept of loud and soft voice by demonstrating the differences.Vocal Marathon: To discuss the concept of freeing the voice via easy voice onset from humming /m/ and/or /h/ initiated words to speech and to increase vocal ability to sustain the voice for 5–10 s, as able.Tag Along Words: To demonstrate the carry-over of skills by generating words from random sounds.What’s Up: To use learned strategies to initiate speech in response to questions.Let’s Face it: To label, identify, imitate, and express emotions. After working on vocal control and speech production, the individuals were able to initiate their voice and answer basic questions using spoken language. The next section utilized vocal control methods to gain knowledge and experience of pragmatic language and increase the ability to use language for different purposes, change language depending on the situation, and engage in conversation.

Pragmatic language work includes targeted skills to build spontaneous verbal communication for conversation. Three primary areas of social pragmatic communication (indicated above) were included in 11 interactive activities with remote access delivered via teletherapy. The goals of each unique online game are described below.

Word Think: To say a word spontaneously without lengthy pausing or hesitationPinpoint: To say what comes to mind spontaneously, take turns speaking, and formulate sentences on a topic.Actor’s Corner—Interactive Scripts: To use intonation and vocal expression, engage in dialogue with turn-taking and topic maintenance, and increase communicative interactions with scripted conversation.Barriers: To increase listening skills, increase direction-giving skills, request clarification, and help reduce possible perfectionistic tendencies.Question Match—Answering Questions: To provide or request information (yes-no and wh-questions); to engage in turn-taking and question-answer routines; and to give sufficient information for listener comprehension and interactions.More Information Please—Changing Questions: To ask questions with intonation and vocal expression; to listen when another person speaks; to take turns asking and answering questions; and to provide a follow-up comment after the other person answers the question.See-Saw—Keep the Conversation Going: To engage in turn-taking during a conversation; to ask/answer yes-no and wh-questions; and to make comments during a conversation (state an opinion or feeling, agree or disagree, provide information or instructions, state an intent, provide clarification, make a prediction, provide a reason, and offer a suggestion).Road Runner—Staying on Topic Track: To maintain a topic or related topics; to stay focused and give information for listener comprehension and interaction; and to engage in a two-way conversation with turn-taking.Conversation Wheelhouse: To attend to stories read aloud with pictures and to share information and interact using the following conversational skills (retelling, questioning, answering, commenting, sharing information, telling a story/relating an event, agreeing, and disagreeing).Conversational Role-Plays: Pragmatic Language: To increase the use of pragmatic language skills found in daily conversations and interactions (greeting, farewells, opening and closing a conversation, showing appreciation, commenting, complimenting, apologizing, requesting clarification or information, stating a problem, making an excuse, complaining, asking for help, offering to help, and providing information).Chat Spin—Informal Conversations: To increase conversational skills by taking turns listening and speaking, keeping the conversation going with questions, answers, and comments; to share experiences, thoughts, and opinions; to use expression in one’s voice; and to smile or act interested in what another person says.

#### 2.4.2. Detailed Case Treatment and Progress

##### Case 1: GB

The following is an overview of GB’s detailed treatment progress.


*GB—Vocal Control*


GB participated in 11 sessions via teletherapy with the initial support of her father, who assisted with setting up the remote connection and remained on standby once each session began. GB was attentive and followed directions despite not using verbalization.

In the first four therapy sessions, the focus was on building foundational knowledge of speech sound production while creating a low-pressure environment. To support her participation, she was allowed to select her preferred mode of communication through teletherapy. GB initially selected the chat feature to relay one-word responses (e.g., “good,” “yes,” “no”). For interactions that required more than one typed response, GB responded to her father, who spoke for her. After each session, the parents received a summary of the session with suggestions for out-of-session practice. 

GB demonstrated an early understanding of the key concepts in Sound Off. She successfully distinguished between oral, nasal, and throat sounds of letters using (English phonemes) and between stops (phonemes that abruptly stop air flow such as /p/ or /d/, for example) and fricatives (/f/ and /s/, for example) and nasal sounds (phonemes that continue in air flow, such as /m/and /n/, for example).

During the middle three sessions, GB independently used the remote-control feature to accurately label variations in pitch and loudness, suggesting growing auditory discrimination skills and comfort with the material (Pitch Pipe and Ramp It Up activities). Using the Vocal Marathon activity, GB required approximately 30 s to begin humming but sustained the hum for only 4 s. GB’s initiation latency decreased to 10 s, and her humming duration increased to 10 s. The introduction of a spinning wheel in activities added an element of expectation and variability, as her response depended on what the spinning wheel revealed once it stopped. GB did not hesitate to embrace this challenge. Her latency fluctuated depending on whether her video was on or off, indicating that visual presence and, possibly, self-consciousness influenced her performance. Soon, GB demonstrated growing independence, engaging fully with both video and sound during teletherapy. She successfully transitioned from sustained humming to vowel initiation and then to producing /m/ initially in words such as “milk,” with her initiation latency improving from 10 s to 6 s. The playful use of the Chicken Scream App v.2.2.3 (voice-activated game) further encouraged sustained voicing while helping GB monitor vocal tension, combining therapeutic goals with interactive and motivating tasks.

Treatment was divided into two sections for these cases; however, one therapist could provide the intervention. The two authors of this paper conducted separate sections based on their areas of specializations. The first, focusing on vocal control, was completed in nine sessions. In the later phase of vocal control therapy, the focus shifted toward expanding GB’s verbal output with Tag Along Words and What’s Up activities. GB independently read her prepared questions to the next therapist, articulating her continued reliance on a humming sound to initiate speech and her awareness of her strategies. GB adapted smoothly to working with the next therapist, requiring no support or prompts from her previous therapist, highlighting her growing ability to initiate voice and generalize vocal control skills with a different therapist. The pragmatic language portion then began. This was continued for an additional 26 sessions.


*GB—Pragmatic Language*


Pragmatic language treatment began with chorally reading with the new therapist. This was a relatively easy task because reading did not require GB to formulate her own speech. To start, the therapist read as the GB whispered simultaneously. This quickly progressed to more audible choral reading at a normal volume, taking turns reading aloud. Spontaneous spoken language was still difficult for GB, and initiating speech was even more challenging. When treatment was initiated, GB took a long time to respond. Hesitations initially ranged from 20 to 53 s to say single words when asked to name a word related to the one on the page. For one example, in the activity Word Think, she was asked to provide a complimentary word for “pencil.” A response could be “eraser,” “pen,” “write,” etc. Informing her that there were no right or wrong answers was beneficial. A timer was used to gauge and reduce the GB’s hesitations. In time, she learned to say phrases such as, “I’m not sure” and “I don’t know.” As treatment progressed, GB formulated sentences using forced-choice questions, giving her a series of two-, three-, and four-word choices, each earning more points toward a motivating reward (pinpoint activity).

An important aspect of her treatment was asking and answering questions. This was part of the Actor’s Corner activity with a script that progressed as one person clicked on the virtual board and read a portion of the dialogue, taking turns to complete words that were missing. Turn-taking was also accomplished during a barriers activity, where one person gave verbal directions to complete a drawing. At the end of the activity, the partners took turns sharing their drawings to see if they matched.

To continue using language for a variety of purposes, treatment focused on the differences between types of questions and the amount of information acquired when asking wh-questions compared to asking yes-no questions (Question Match activity). Intonation was also a focus in the interactive activities to help engage her communication partner (More Information Please activity). It is important to focus on strategies to keep conversations going. The therapist began by typing a sentence, which was either a question or a comment. Each person used a mouse click to take their turn and color-coded the sentence as either a question, response, or comment (See-Saw activity). Given this support, GB learned to increase the amount of information that she shared. Initially, her responses were limited to one to three words in the form of answers. Commenting is a new and challenging task. With skill building and practice, her responses averaged five words, with several questions and answers that included initiating. GB gained knowledge about how to extend a topic and move to related topics, whereas prior to this work, her communication was similar to someone being interviewed, with the communication partner asking questions and GB responding with as few words as possible. Using a visual interactive track as part of an online game, the goal was for the GB to stay out of the off-topic zone. The more she stayed on a topic or a related topic, the more on track she remained within the online game board (Road Runner activity).

GB also worked on retelling information, retelling a story, sharing information about an event, agreeing, and disagreeing. Each skill was practiced using a photo to depict an event along with a problem to be solved. With the photos and accompanying scenarios, new communication opportunities were presented for practice (Conversation Wheelhouse activity). GB’s pragmatic language skills increased through the Conversational Role-Plays activity, as she practiced saying hello and goodbye, thanking someone, complimenting someone, apologizing, requesting clarification, stating a problem, making a complaint, asking for or offering help, and providing information. GB said that she was most comfortable greeting and apologizing, but least comfortable asking for help and saying goodbye. She noted that it was the hardest to say “Thanks” to teachers at the end of class. All these skills were accomplished with the addition of informal conversations, guessing games, and outreach practice. The Chat Spin activity gave GB an opportunity to speak spontaneously about a topic she landed on using a spinning wheel. It was helpful for GB to think about what she knew about topics (pets, gifts, vacations, music, school, etc.) and what she wanted to learn. It was important to talk with GB about her fears and feelings of responsibility for others’ feelings. GB began to overcome some of her challenges, reducing her worries about pleasing people and trying to speak perfectly. Conversations became easier for GB during the course of treatment. Her self-rated accomplishments in using language for different purposes increased from approximately 47% to 96% on the Social Communication Skills: Pragmatics Checklist ([7]). These included greeting someone when entering a room, saying goodbye when leaving, thanking someone, making comments, complimenting someone, apologizing, requesting information, stating a problem, complaining, asking for help, offering help, and providing the needed information.

To determine what made certain communication situations difficult for GB, ratings from 1 (least) to 5 (most) were used. Saying hello to friends at dance class was challenging (rated 4). When GB could not figure out what made one of her challenges a high rating, she was given choices to determine how realistic the situation was. She realized that the most likely reason she had not greeted the girls in her dance class was that it was unfamiliar, something she had not done before. This made the task frightening and something she worried about. She was concerned about what the girls (who had never heard her speak) would think or say. In therapy, visualizing actually doing the task and role-playing were helpful. Repeating back what someone said to her was also practiced. She learned to ask the same question in return to the person who asked her first. Often this is a good strategy for questions like “How are you?” or “What did you think of ___________?”

Throughout the treatment, outreach work was important for making progress. GB received guidance by calling store clerks and asking questions about their hours, location, and products. After practicing together, GB called a store when she was alone. Each week, she chose a new challenge that was modeled by watching the therapist do it in a live situation and then role-played in therapy to repeat by herself. Each goal was supported by GB’s parents, who received progress notes after every treatment session. At school, GB increased her ability to ask questions of her teachers. At home, she ordered pizza for her family over the phone. She also began ordering ice cream at a store in person. Saying ‘hi’ to a girl in her dance group was a major accomplishment that took months, but she did it!

As part of the treatment, GB became acquainted with other people who had SM. The therapist helped her create questions of interest related to her journey with SM. GB asked five questions using speech to interview two young adults online who had overcome their SM. Her questions to the young adults who had overcome SM were as follows: (1) How old were you when you started having trouble getting your words out? (2) What was it like for you in school to have this experience? (3) What did you do to overcome this problem? (4) Do you still have trouble getting your words out? (5) What do you want to do in the future? This experience was profound, and according to her parents, it had a significant impact on her understanding of SM and her belief that she could overcome SM. Toward the end of treatment, GB said she was proud of herself and what she accomplished. She was especially pleased with her school presentation, where she presented her side of a debate in front of her class, asked a question at a school visitation about a prospective high school (with her parents there), ordered dinner on the phone, answered the doctor who asked her several questions at a well-check-up visit, asked a sales person a question at a bookstore, answered her teacher’s questions in the school hallway, and read in front of her class. Relaxation techniques included visual imagery of success and positive affirmations to help her reduce her worry about making errors.

Motivational rewards and outreach activities were key to the GB’s ongoing progress. Her motivational rewards were often related to activities with her family. She chose to play board games, watch movies, buy ingredients to cook dinner together, go bowling, play miniature golf, and engage in other family activities.

##### Case 2: KD

*The following provides an overview of the KD treatment progress*.


*KD—Vocal Control*


KD attended nine teletherapy sessions with the initial support of his mother, who assisted with setting up the remote connection but stepped away once each session began. Initially, KD participated via audio only because his laptop did not have a camera. Despite the lack of visual input, KD consistently responded verbally, using his voice to answer and communicate basic needs, such as requesting remote control access, although he did not typically initiate conversations. His mother received a summary of each session with details for out-of-session practice.

The initial session focused on building rapport and establishing the expectations. Conversations and simple auditory tasks were used to ease the KD into the therapeutic environment. Recognizing his interest in video games, the therapist incorporated visual-spatial games that KD controlled from his computer, allowing for an interactive and motivating experience for him. KD’s voice during this session was noted to be high-pitched, tense, and of low volume compared to age expectations. A slight improvement in vocal quality was observed by the end of the session, as he became more comfortable.

KD was on time at the start of each subsequent session, with his mother present for technical support. Other sessions followed a similar pattern, beginning with an icebreaker game like the “Parking Lot Game” which progressed in difficulty (Levels 1 and 2 during session two and Level 3 during session four). The game required the player to move a car out of a parking lot by strategically shifting other obstructing cars. The game required the player to move a car out of a parking lot by strategically shifting other obstructing cars. Based on his interest, KD demonstrated strong technological control and completed tasks with minimal hesitation. Using the activities as an icebreaker helped KD build rapport and ease his engagement in speech-related tasks.

Each session reinforced foundational speech concepts. KD was introduced to sound placement and distinguished between oral, nasal, and throat-based sounds. Using interactive PowerPoint slides, he promptly responded to questions regarding sound placement, exhibiting a good understanding. Other activities were expanded into interactive categorization activities, where KD successfully classified sounds as voiced or voiceless, nasal or oral, and stop or continuous. He not only responded correctly but also began to self-monitor his performance and rehearse the sounds independently, signaling emerging self-awareness in his speech production (Sound Off activity).

The complexity of the tasks increased with the introduction of the pitch and loudness concepts. Auditory samples with varying pitches (high, normal, and low) and loudness (whisper, soft, normal, and loud) were presented. KD accurately labeled the samples using interactive features, demonstrating a solid perceptual understanding (Pitch Pipe and Ramp It Up activities). However, when tasked with imitating the variations he heard, KD faced significant challenges. His own vocal productions remained flat, maintaining a consistent pitch and loudness at his habitual speaking level. During imitation tasks, KD exhibited clear signs of laryngeal tension, producing pitch breaks, maintaining phonation for only short bursts (approximately 2 s), and struggling to shift to softer or whispered voices. The Vocal Marathon activity was used to help him gain these skills. When asked if he realized the difficulty he was having, KD expressed genuine surprise, indicating that these underlying vocal issues were not within his awareness before therapy.

The sessions extended the previous week’s goals by focusing on reducing vocal tension and increasing vocal endurance through prolonged humming with the Vocal Marathon activity. KD successfully sustained humming for over 10 s with a notable reduction in laryngeal tension. The session progressed to incorporating /h/-initiated words such as “hello,” providing a bridge to speech through the coordination of airflow. KD showed increased confidence and smoother transitions in voice onset, reflecting the early generalization of vocal control. KD continued with the Tag Along Words activity and demonstrated carry-over of skills by generating words from random sounds. This led to greater ease in answering open-ended Wh-questions in the What’s Up? activity. A randomized Q&A game using numbered blocks was used to elicit spontaneous speech and handle unpredictable prompts. KD self-corrected, responded to the feedback, and completed all the assigned tasks. His participation was strong in structured activities, although his responses were more variable during open-ended prompts. KD demonstrated consistent improvement in vocal initiation, voice quality, and sustained phonation across tasks, indicating increasing stability in voice use. He was still hesitant to spontaneously ask questions or engage in conversations. Throughout, KD’s affect was flat. Therefore, the Let’s Face It activity was attempted to increase the identification of emotions from pictures and statements, as well as the imitation of emotions based on the sample provided. KD performed well on all tasks until this point. However, he exhibited difficulties when asked to verbally produce an emotion for the therapist to guess. He asked, “What do I say?” He required cues like: “How would you feel if you got a new video game? Once he said “happy” then the clinician asked him to say “I just got a new game” in a way that conveyed he was happy. KD acknowledged that the task was more difficult. The therapist let him know that not all activities would be easy the first time, and with practice, he would do better, just like he does with practice on his new video games. Although attempts were made, his intonation remained monotonous.

The final vocal control session focused on reviewing the progress and consolidating skills. A second therapist, specializing in pragmatic language treatment, joined the session. After initial hesitation, KD began engaging with the new therapist by asking scripted questions. KD noted how far he had come with treatment as he answered the yes-no questions about his progress. The therapist guided the patient through a mix of structured vocal tasks and open-ended questions. KD stated a few personal goals for communication and named his favorite activities, which included: His voice was clear and seemingly confident, indicating that he was ready to move to the next level. KD’s smooth initiation and flow with known material, despite hesitation with abstract or imaginative questions, demonstrated readiness for more complex tasks to generalize vocal control into pragmatic language use. This was continued for an additional 30 sessions.


*KD—Pragmatic Language*


KD was punctual and attentive during the online sessions. Initially, he worked with his camera off and had vocal tension, struggling to sustain the vowel sounds for more than two seconds. After working on vocal control, he could sustain a vowel sound for 15 s and vary his pitch and loudness with minimal difficulty. KD answered questions in complete sentences without hesitation once his camera was on, but did not initiate communication or engage in conversation. Social pragmatic communication work focuses on increasing engagement and spontaneous speech with longer sentences.

KD responded to each guided activity in the pragmatic language module. He worked best when the activities were realistic and meaningful to him. He needed to see the relevance of the treatment. Using the Word Think activity, he learned about the concept of spontaneity. To help him think of a related word, he was taught to visualize an image of the given word. He helped develop a hierarchy for easy-to-difficult speaking tasks. He learned that outreach would begin with easier tasks, such as answering questions, and progress to medium-difficulty tasks, such as asking for something at a store or saying thank you to someone at the pharmacy. Difficult tasks included asking teachers for information and ordering food over the phone. To help reduce his hesitations within the pinpoint activity, he created sentences with the random word game. At the end of this session, he said goodbye and waved, showing progress. With his dog nearby during the session, he smiled too.

The Actor’s Corner activity provided KD with written scripts about situations that are common in high schools. During this turn-taking activity, he replied with a few words using sentence completion tasks. He rated his ability to plan what to say next (from 1 for easy to 10 for very difficult) at an ‘8.’ He said it did not really matter because he spoke only when necessary. During the next activity that required following and giving directions, he remained engaged and gave the therapist 10 instructions (barriers activity). For the Question Match activity, KD asked and answered 36 questions as he took turns using a set of question cards. He initiated his own questions, asking about the pictures on the wall of the teletherapy room.

More Information Please was a valuable activity for showing KD how information changes based on what someone asks. Wh-questions and Yes-No questions were compared. For example, “When do you go to sleep?” and “Do you go to sleep early?” offer different types of information. This was an area he caught on to quickly, with the ability to change questions from one type to another.

During the next online session, KD and the therapist met a young adult who had SM throughout their childhood. During a previous session, KD worked with the therapist to create five questions to ask this individual about SM. The questions were as follows: (1) How old were you when you started having trouble getting your words out? (2) What was it like for you in school to have this experience? (3) What did you do to overcome this problem? (4) Do you still have trouble speaking? (5) What do you want to do in the future? Although he was apprehensive about asking the five questions when KD and the guest met online, he read them aloud, and the interview went well. However, there was no interaction other than asking the questions. Knowing that others have improved in their ability to speak appeared to be of interest to KD. According to his mother, in a follow-up email, she thought that the meeting offered KD some encouragement. Interestingly, KD told his mother that, “the guy had it worse than me.” When asked about overcoming SM, KD said he did not care and that he talked when he needed to and did not think it was a problem.

The concept of affect has been mentioned. Given five small Kimochi-feeling-pillow faces, the KD was asked to choose the one that most resembled him. The choices were happy, sad, brave/proud, mad, and left out/concerned. He picked left out/concerned. The therapist was pleased that he shared a meaningful thought about his life, even though he did not elaborate. More emotional words were heard at home, as KD’s mother mentioned that KD had apologized to her during the week.

The See-Saw activity helped KD analyze sentences and use various yes-no questions, wh-questions, answers, and comments. Using the type-and-talk activity was not his favorite as he had to come up with sentences to keep the conversation going. Again, he said he would talk if he needed to and that he did not see a point in speaking if it was not needed for a specific purpose. This fact was inaccurate, according to his mother. She conveyed that even when there was a need to talk, such as at the supermarket to find an item, he did not do so and left the store without the item.

In the Conversation Wheelhouse activity, KD worked on pragmatic speech acts for disagreeing, commenting, asking questions, and retelling. With practice, he had a 73% success rate with 14 opportunities. KDs most frequent pragmatic language use was saying ‘bye’ and ‘thank you.’ He rarely said hello, made comments or gave compliments. He also rarely apologized, asked for help, or made excuses. The Conversational Role-Play activity provided practice for a variety of pragmatic language skills found in daily conversations. KD was reluctant to engage because he said the scenarios were not ones he had actually encountered in his life. However, he did practice working on topics with which he could identify, such as stating a problem, commenting, offering help, providing information, and showing appreciation. KD did not like engaging in verbal role-play scenarios, but he attempted seven of ten interactive scenes with spoken sentences.

Throughout this work, outreach improved. KD ordered pizza for delivery at home. He also engaged in some reciprocal communication and made comments when reminded with, “What do you think?” KD’s self-rating from 1 (sad) to 10 (happy) reached ‘7’ during this point in treatment, a definite improvement. The final activity, Chat Spin¸ was a success as KD and his mother played the activity with the therapist, gaining points and rewards for guessing answers based on accurate knowledge about the other person.

According to his mother, KD began responding more at school and even said ‘thank you’ to a person who held the door open for him at a convenience store. KD was interested in learning to drive a car. Because this was a practical need, it was incorporated into the therapy. He worked on answering questions from a driver’s test manual. Items he did not know led to questions and discussions about problem solving.

## 3. Results

### 3.1. Case 1—GB’s Treatment Outcomes

The treatment included 35 sessions over 16 months. Pauses in treatment were related to vacations and time off due to other school, family, and extracurricular responsibilities. Throughout the sessions, the GB showed increasing autonomy, comfort, and vocal participation in teletherapy. This steady progress indicates significant behavioral, communicative, and emotional growth.

Results from the *Selective Mutism Questionnaire* (SMQ) ([3]) showed GB’s progress from her parents’ and her perspective. The SMQ has a total of 51 points possible based on 17 items, each with a possibility of 3 points for full frequency of speaking in a given situation (17 × 3 = 51). Using the SMQ rating scale at the start of therapy, GB’s frequency of speech at school, with family, and in social situations (according to her parents) was 18/51 (35.3%), and after therapy, her SMQ score was 44/51 (86.3%). According to GB, she assigned herself a final score of 43 out of 51 points (84.3%), indicating substantial progress. See Table 1.

On the *Social Communication Skills—Pragmatics Checklist* ([7]), GB had a pre-treatment score of 63/135 (46.7%) for meeting pragmatic language objectives for communication. After therapy, her pragmatic checklist score was 130/135 (96.3%). The areas included specific examples of stating needs, giving directions, expressing feelings, interacting with a variety of pragmatic speech acts, asking questions, and sharing knowledge and imagination. Gains were observed in all areas. The most significant changes were observed in her ability to express feelings, offer an opinion, use appropriate social rules for greetings and farewells, attend to the speaker, initiate a topic of conversation, make an apology, request clarification, ask questions to gather information, and tell a story. See Table 2.

GB’s therapeutic journey for vocal control illustrated significant and measurable growth in both her communication abilities and her emotional readiness to participate. Through treatment, GB moved from a place of high anxiety and limited output to confident, independent speech participation, positioning her well for continued success in future therapy endeavors.

Several key themes emerged from GB’s therapeutic journey. Her progression from nonverbal, father-mediated communication to independent verbal participation marked significant growth in trust, confidence, and desensitization to communication pressure. The use of humming proved to be an essential bridging strategy throughout her therapy, reliably supporting voice initiation and tension management, even as tasks became increasingly complex. Across sessions, GB showed clear, measurable reductions in latency of responding from an initial 30 s to an average of 6 s, indicating not only therapeutic success but also voice activation and initiation. Additionally, her increasing confidence and autonomy, especially during the final sessions, were critical indicators of her readiness for more advanced language tasks and for long-term communicative success.

GB’s smooth transition to a new therapist, with maintained engagement and independence, spoke to the strength of the therapeutic rapport established and her ability to manage change, which is a crucial component for maintaining momentum in therapy. Areas of growth were observed across school, family, and social settings. When GB started treatment, she did not speak to her teachers or the school staff. She spoke to selected peers at school. At the final points of treatment, she consistently spoke to her teachers, even when called upon in front of the class. At home, she did not speak to her relatives, including her grandparents and family friends. By the end of the treatment, she often spoke to all her relatives and family friends. Regarding the frequency of speech in social situations outside school, GB seldom spoke during extracurricular activities or to clerks or waiters. By the end of therapy, she consistently spoke during extracurricular activities and with store clerks and waiters.

GB’s parents said, “We had GB see many professionals, and she was even put on medication at one time, but this has been amazing, what you have done for her to speak. Your two-prong approach, getting the voice started and then the language approach, has been making all the difference.” GB’s teachers, grandmother, and physician also noted positive changes.

### 3.2. Case 2—KD’s Treatment Outcomes

The treatment included 39 sessions over 12 months. Missed sessions were related to appointments, time off due to family vacations, college visits, and graduation. Throughout the sessions, KD showed increasing confidence and participated in outreach activities.

The SMQ from the start to end of treatment indicated KD’s progress in the frequency of speaking at school, with family, and in social situations outside of school. Table 3 shows the progress over time in all three settings.

The *Social Communication Skills Pragmatics Checklist* was completed by KD’s mother with guidance from the speech-language pathologist (SLP) at pretest and post-test times. The improvements by category are presented in Table 4.

KD’s progression through the initial part of treatment for vocal control demonstrated an increase in self-confidence, communication skills, technological comfort, and therapeutic engagement. Initially, KD participated only through audio, which provided a consistent and reliable communication channel. This stability was crucial for maintaining therapeutic continuity and allowing verbal tasks to progress seamlessly, regardless of video availability. KD transitioned to full visual interaction to enhance engagement through facial expressions and gestures with modeling. These tools are essential for building communicative competence and fostering rapport. KD showed persistence and engagement despite his difficulties with intonation and complex sentence structure. Activities were carefully selected to maintain his motivation while gradually introducing more complex voice production skills. His awareness of his own voice grew, laying an important foundation for further therapeutic work targeting pitch modulation, loudness control, and vocal tension reduction.

Visual-spatial games play a pivotal role in engagement. These icebreakers supported problem-solving, encouraged flexible thinking, and helped KD ease into the session’s environment. The interactive nature of these games provides an indirect path to achieving therapeutic goals.

A gradual but distinct transition occurred from active parental involvement in the early sessions to a more passive supportive role later on. This shift illustrated KD’s growing independence and increasing comfort in navigating the virtual format. It also highlighted KD’s ability to manage session routines and expectations with minimal external assistance, which is an important step toward self-regulation and therapeutic ownership. By the end of the vocal control sessions, KD had established greater independence, deeper engagement, and increased comfort with the therapeutic process.

KD accomplished several outreach tasks by the end of treatment. These included ordering for himself at restaurants, responding to questions with basic information, answering a phone call and noting information to tell his parents, saying thank you to people, placing an order at a fast food establishment, leaving a message at a pharmacy to find out about a vaccine, telling a neighbor he was going to college, apologizing to his mother, answering questions from a clerk at the Office of Vocational Rehabilitation, responding to a veterinarian who asked him questions about his dog, and earning his driver’s license, which required talking to the driving evaluator.

Toward the end of treatment, KD’s mother wrote, “I’m sure you know this, but I just want to express my appreciation of your work with my son. I can’t tell you how even baby steps in progress are encouraging in what has been a long road and will continue to be a long road. He made more progress in one year’s time than we’ve experienced over several years. Maybe one day he’ll speak in front of your class!”

At the end of his 12th grade school year, KD was accepted into college to study computer cybersecurity. According to a follow-up phone call with his mother, KD was living on his own, had a roommate, and went to class regularly. He was navigating college life independently.

## 4. Discussion and Conclusions

The vocal control element of the protocol contributes a novel, physiologically informed framework to the literature on SM. It presents voice initiation as both a neuromotor task and a psychological milestone. The vocal control-pragmatic language approach: (a) targets physiological inhibition, not just behavioral avoidance, (b) incorporates a neurobiological understanding of anxiety and vocal tension that can inhibit vocal cord vibration necessary for voice and speech production, and (c) introduces vocal control training as a foundational step, supported by objective physiological data via sEMG. By doing so, it integrates brain-based science, vocal physiology, and therapeutic sequencing into one coherent model, broadening the scope of SM treatment and enhancing its efficacy. Following the ability to initiate voice for speech, pragmatic language skills must be taught. Individuals who have not spoken in a variety of communicative situations often lack the skills necessary for the social use of language. This includes adapting language to a variety of situations, people, and social contexts, including turn-taking, topic maintenance, and conversation, to name a few.

The therapeutic journeys of GB and KD provide valuable insights into the effective treatment of SM. Their progress underscores the importance of therapy using an approach that integrates vocal control and pragmatic language. By fostering confidence in speaking and enhancing social communication skills in sessions and through outreach work, therapy can improve the quality of life of children with SM.

GB’s progression through therapy underscores the effectiveness of a multifaceted approach that combines vocal control and pragmatic language work. Her initial reluctance and significant latency in initiating speech improved markedly, demonstrating the importance of practice with gradual exposure to increasingly challenging communication situations. The vocal control module facilitated GB’s ability to initiate her voice with reduced vocal tension, while the pragmatic language module helped her use language for various communicative intents. The clinical implications of GB highlight the necessity of a tailored intervention plan that addresses both the physiological and psychological aspects of SM, with a focus on speech and language. The integration of vocal control exercises, such as humming and whispering, alongside pragmatic language activities that highlight the use of language for specific purposes, fostered confidence in speaking and enhanced their social communication skills. GB’s ability to generalize these skills with practice across different settings and interactions indicates her potential for long-term communicative success.

KD’s journey through therapy illustrates the challenges and successes of treating SM, particularly in the context of comorbid conditions like autism. His progression from nonverbal participation to full engagement in therapy sessions highlights the importance of creating a low-pressure and motivating environment. The use of interactive activities and visual-spatial games was helpful in engaging KD and easing him into vocal tasks. The clinical implications of KD emphasize the need for individualized treatment plans that consider the unique challenges posed by coexisting conditions. The gradual increase in task complexity, combined with consistent parental support and structured therapeutic routines, contributed to KD’s communicative growth. His ability to manage session logistics independently and engage in verbal interactions, seeing a purpose for speaking in situations, indicates a promising trajectory for continued development in social communication.

The case reports reflect the clinical presentation, management, and outcomes of two individuals, which limits the generalizability of our findings. Case reports do not represent broader population trends. Clinical observations may be influenced by subjective factors. While every effort was made to provide an accurate and comprehensive account of GB and KD treatment outcomes, the absence of a control group limits the strength of our conclusions.

The following seven points highlight the essential factors in supporting SM treatment in teenagers using a teletherapy platform.

Early Detection and Intervention: Early recognition of SM and timely intervention are critical in preventing long-term anxiety disorders and communication impairments.Multifaceted Therapeutic Approach: An integrated treatment plan addressing both vocal control and pragmatic language skills is essential for effective therapy.Individualized Treatment Plans: Tailoring interventions to the unique needs and abilities of each child, especially those with comorbid conditions, enhances therapeutic outcomes.Gradual Exposure and Outreach: Gradual exposure to challenging communicative situations fosters confidence and reduces anxiety.Parental Support and Engagement: Consistent involvement and support from parents are crucial in reinforcing therapeutic gains.Generalization of Skills: Encouraging the generalization of communicative skills across different settings and interactions is key to long-term success.Motivating and Engaging Activities: Strategic rewards for accomplishing interactive, meaningful activities that resonate with the child’s interests, enhance engagement, and help facilitate skill acquisition.

### Limitations, Barriers, and Future Directions

Case reports offer valuable clinical perspectives; however, there are limitations that restrict their broader applicability. The reliance on two individual case studies limits the generalizability of the findings. The results observed in GB and KD may not be representative of the wider population of children and adolescents with SM, although they show various progress profiles. The cases primarily focus on teenagers. There is a need to explore the efficacy of this vocal control-pragmatic language intervention across a broader age range, including younger children (beginning with those in upper elementary grades) and older adolescents (up to 21 years of age). Additionally, without a control or comparison group, it is difficult to attribute improvements solely to the intervention, as spontaneous improvement or external factors cannot be excluded. Clinical observations and outcomes in case reports are susceptible to subjective bias, as they may reflect the interpretations and expectations of the clinician or client. While progress has been documented for more than a year, long-term outcomes and maintenance of communicative gains remain unverified over the long term.Families seeking effective therapeutic interventions for children with SM face various barriers that can limit treatment success. For young children, developmental readiness may be of concern, as short attention spans and limited comprehension can reduce the effectiveness of structured therapy, especially when delivered virtually. Although technology provides opportunities for remote access, it may not be available to everyone. Not all families have reliable Internet, appropriate devices, or the skills needed to fully engage in virtual treatment. Family involvement is a critical factor. Inconsistent support at home, combined with cultural or language mismatches between families and providers, can weaken the treatment outcomes. Children themselves vary widely in their needs and responsiveness. Those with co-occurring conditions, such as autism or anxiety, may require alternative approaches. External barriers, such as the limited availability of qualified specialists, insurance restrictions, and scheduling conflicts, can make access to care even more challenging. Additionally, the skills and experience of the therapist are essential. Together, these challenges highlight the need for flexible and personalized treatment plans. Successful interventions must consider each child’s developmental level, home environment, cultural background, and emotional readiness.By systematically addressing these limitations and barriers, future work may provide a comprehensive and reliable understanding of effective long-term SM interventions. This will ultimately inform best practices and improve outcomes for children, teenagers, and families navigating SM.

## Figures and Tables

**Figure 1 behavsci-15-01013-f001:**
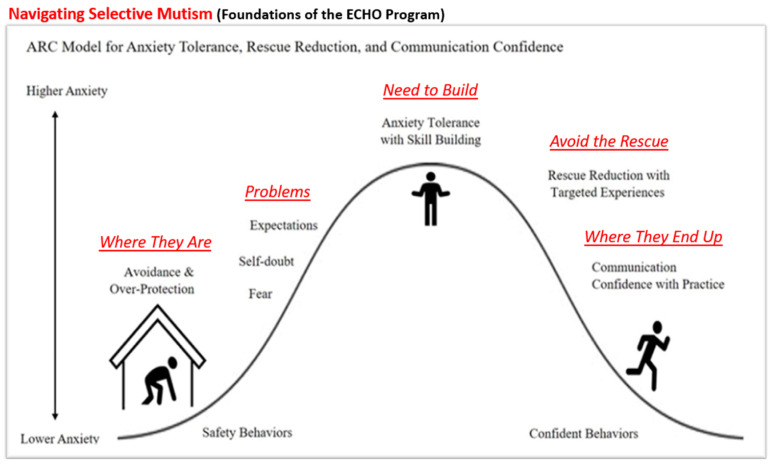
ARC Model Foundations of Reducing SM ([11]).

**Table 1 behavsci-15-01013-t001:** GB’s Pre-to-Post SMQ Scores.

SMQ Settings	Pretest Completed by Mother	Post-Test Completed by Mother	Post-Test Completed by GB
School (3-point scale.)	1.33 (44.3%)	2.83 (94.3%)	2.5 (83.3)
Family (3-point scale)	1.4 (46.7%)	2.83 (94.3%)	2.67 (89%)
Social Situations (3-point scale)	0.6 (20%)	2.0 (66.7%)	2.4 (80%)
Total Points (based on 17 items 3 points possible for each = 51	18/51 (35.3%)	44/51 (86.3%)	43/51 (84.3)

**Table 2 behavsci-15-01013-t002:** GB’s Pre-to-Post Pragmatic Checklist Scores.

Pragmatic Categories	Pre (Completed by Parents)	Post (Completed by Parents)
Stating Needs	8/15 (53%)	15/15 (100%)
Giving Directions	5/9 (56%)	8/9 (89%)
Expressing Feelings	15/21 (71%)	21/21 (100%)
Interacting	17/45 (38%)	43/45 (96%)
Asking Questions	7/15 (47%)	14/15 (93%)
Sharing Knowledge/Thoughts	11/30 (37%)	29/30 (97%)
Total	63/135 (46.7%)	130/135 (96.3%)

**Table 3 behavsci-15-01013-t003:** KD’s Pre-to-Post SMQ Scores.

SMQ Settings	Pretest Completed by Mother	Post-Test Completed by Mother	Post-Test Completed by KD
At School	1.33 (44.3%)	1.83 (61%)	1.5 (50%)
With Family	0.66 (22%)	0.83 (33.3%)	2 (66.7%)
In Social Situations	0.2 (6.7%)	0.6 (20%)	1.8 (60%)
Total Points	13/51 (25.5%)	19/51 (37.3%)	28/51 (54.9%)

**Table 4 behavsci-15-01013-t004:** KD’s Pre-to-Post-Pragmatic Checklist Scores.

Pragmatic Categories	Pre (Completed by Mother)	Post (Completed by Mother)
Stating Needs	6/15 (40%)	11/15 (73%)
Giving Directions	0/9 (0%)	3/9 (33%)
Expressing Feelings	2/21 (9.5%)	7/21 (33%)
Interacting	2/45 (4%)	21/45 (46%)
Asking Questions	0/15 (0%)	2/15 (13%)
Sharing Knowledge/Thoughts	8/30 (26%)	10/30 (33%)
Total	18/135 (13.3%)	54/135 (40%)

## Data Availability

The original contributions presented in this study are included in the article. Further inquiries can be directed to the corresponding author.

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
