# Peer review of "A Physiological Approach to Vocalization and Expanding Spoken Language for Adolescents with Selective Mutism"

_behavsci, 2025, doi:10.3390/bs15081013_

Round 1

Reviewer 1 Report

Comments and Suggestions for Authors

Thank you for the opportunity to review this manuscript. The authors bring an exciting perspective to the treatment of teens with selective mutism, incorporating vocal control tasks and pragmatic communication tasks alongside more traditional strategies. There are some improvements I recommend below to improve the manuscript. 

  1. In Participants:
    1. Capitalize the written form of Selective Mutism Association 
    2. Can you also clarify what it means that the families interviewed the SLP and psychologist or was it supposed to say the families were interviewed by the SLP and psychologist?
  2. Duration - make the duration of the treatment in the format for KD as it is more clearly stated. It is hard to know for GB if the 35 is weeks or sessions? Make this consistent across the 2 cases in terms of how it's written.
  3. For KD - you have him listed as 16, but then noted in medical history about him being the age of 17 when discussing bone age. Clarify the accurate age. Also, please clarify the statement about all medications being discontinued because of being ineffective or due to his bone age being reduced? This is not clear.
  4. It would be helpful when describing the child’s success/progress with the tasks to comment on how many sessions this took, or some capture of timeline per module in the protocol. It may be helpful to put in a table that highlights how many sessions/weeks it took to move from one module to the next for each case. 
  5. For GB - why was there a new therapist? Was this a part of the treatment protocol or just by chance?
  6. There is redundancy about the parents receiving session progress notes - could describe it the second time "as they were assigned out of session practice".
  7. For KD, you describe the parking lot game and his difficulty with progression in this task, but it is unclear how this task is related to speech or pragmatic communication. Please clarify how this relates.
  8. For both cases in the results section, I recommend moving the data to start the section with SMQ and pragmatics checklist, followed by more anecdotal descriptions of change. For GB, make the smq table the same format as the pragmatic checklist table and conistent with how its reported for KD, instead of explaining the sum and total 3 different times in the manuscript. It also does not need to be described again how to score the SMQ in the results section, given you describe it in the methods section.
  9. KD results:
    1. Line 861 SMQ can be used given it has already been abbreviated several times before.
    2. Do not need to explain the pragmatic checklist scoring again - can just table to explain improvements. In the table for pragmatic checklist - make the totals rows even.
    3. Also, you mention that the SLP completed the Pragmatics Checklist two times, but this data is not presented or explained in text. What did the SLP report?
  10. The conclusions do not identify how using the vocal control element of the protocol adds to the literature and makes these findings unique. The main points they highlight about teletherapy with teens are more general points without highlighting the importance of the vocal control elements discussed thoroughly in the intro and methods. 

Reviewer 2 Report

Comments and Suggestions for Authors

Review of Manuscript ID: behavsci-3674897

This was a case study report examining treatment for two adolescents with SM. Overall, the paper was clear and informative. I have minor comments below for the authors to address in a revision.

  • Page 19 – the authors note that KD’s parent and speech language pathologist completed the Social Communication Skills Pragmatics Checklist but I only see the results from KD’s parent in Table 4. Please also add in the results from the SLP for completeness.

  • The authors rightfully discuss that lack of a control group and a small sample size are limitations. I think it is also worth noting that the sample contained only adolescents (not children). The title of the paper indicates “children” which does not seem to match the age of the sample. I recommend that the authors expand on the limitations and future directions section, for example, including methods to ensure the effectiveness of the treatment (e.g., larger sample, control group, wider age range, demographics, etc).

  • Related to the point above, in the discussion, could the authors speak a bit about barriers to treatment effectiveness, either anecdotally or perceived barriers? (e.g., what is the youngest age it is used; requires access to computer)

Round 2

Reviewer 2 Report

Comments and Suggestions for Authors

The authors did a wonderful job addressing my previous concerns!